

# Dragon boat exercise reshapes the temporal-spatial dynamics of the brain

Hongke Jiang[1], Shanguang Zhao[1], Qianqian Wu[2], Yingying Cao[2], Wu Zhou[2], Youwu Gong[1], Changzhuan Shao[1] and Aiping Chi[2]

[1] Department of Physical Education, Shanghai Maritime University, Shanghai, China
[2] School of Physical Education, Shaanxi Normal University, Xian, China

## ABSTRACT

Although exercise training has been shown to enhance neurological function, there is a shortage of research on how exercise training affects the temporal-spatial synchronization properties of functional networks, which are crucial to the neurological system. This study recruited 23 professional and 24 amateur dragon boat racers to perform simulated paddling on ergometers while recording EEG. The spatiotemporal dynamics of the brain were analyzed using microstates and omega complexity. Temporal dynamics results showed that microstate D, which is associated with attentional networks, appeared significantly altered, with significantly higher duration, occurrence, and coverage in the professional group than in the amateur group. The transition probabilities of microstate D exhibited a similar pattern. The spatial dynamics results showed the professional group had lower brain complexity than the amateur group, with a significant decrease in omega complexity in the α (8–12 Hz) and β (13–30 Hz) bands. Dragon boat training may strengthen the attentive network and reduce the complexity of the brain. This study provides evidence that dragon boat exercise improves the efficiency of the cerebral functional networks on a spatiotemporal scale.

## INTRODUCTION

Moderate exercise is shown to boost cognitive abilities, such as memory and attention, and reduce the risk of neurological issues; an understanding of the importance of exercise for brain functional networks is growing in the fields of neuroscience and sports medicine (*Slomski, 2019*). The brain's functional state involves multiple neurophysiological features (*Herold et al., 2019*; *Voss et al., 2010*), but the spatial and temporal characteristics of how brain networks change after exercise are still unknown.

Electroencephalography (EEG) offers a valuable tool for studying the brain dynamics. In recent years, EEG microstates have emerged as a promising neurophysiological tool and have been widely used to observe and evaluate the dynamic features of brain networks in the millisecond time dimension (*de Bock et al., 2020*). These microstates of the brain, also called the "atoms of thought," have a rich set of potential neurophysiological parameters in their time series that closely reflect the temporal and spatial characteristics of the brain's electrical activity (*Britz, Van De Ville & Michel, 2010*). EEG microstates, characterized by

Corresponding author
Shanguang Zhao,
sgzhao@shmtu.edu.cn

brief periods of stable scalp potential topographies, are thought to reflect the functional states of large-scale brain networks (*Khanna et al., 2015*). These microstates are typically classified into four main classes, labeled A–D, each with distinct spatial and temporal characteristics (*Koenig et al., 2002*). Microstate A is associated with resting-state networks and default mode activity, reflecting introspective processes and mind wandering (*Custo et al., 2017*). Microstate B is often linked to visual processing and sensory perception (*Van de Ville, Britz & Michel, 2010*), while microstate C is related to attentional processes and cognitive control (*Custo et al., 2017*). Microstate D has been implicated in higher-order cognitive functions, including memory encoding and retrieval (*Van de Ville, Britz & Michel, 2010*). Microstate features can be described using parameters such as duration, occurrence rate, contribution, and transition probability, reflecting the level of microstate transitions and the stability and activation trend of the underlying neuro components in different microstate classes (*Kim et al., 2021*). Research has shown that microstate abnormalities are closely related to the pathological processes of various cognitive function disorders, such as Alzheimer's disease (*Chu et al., 2020*), schizophrenia classification (*da Cruz et al., 2020*), and depression (*Zhao et al., 2022*). However, the correlation between exercise and EEG microstate changes is currently underexplored. *Spring, Tomescu & Barral (2017)* found that after acute exercise, the duration and stability of microstate C increased and the transition probability from microstates B and D to microstate C increased. *Gu et al. (2022)* showed that in resting state, the duration, occurrence rate, and coverage rate of microstate D in professional sharpshooters were significantly higher than those in amateur sharpshooters, and the likelihood of transition from other microstates to microstate D was the highest. A recent study demonstrated that playing team sports increased the duration of microstate C in depressive college students and improved the transition probability from microstate B to microstate D (*Liang et al., 2021*).

EEG rhythm with different frequency bands reflect distinct neural processes (*Başar et al., 1999*). Delta waves (0.5–4 Hz) are associated with deep sleep and cognitive processes such as memory consolidation (*Bernardi et al., 2019*). Theta oscillations (4–8 Hz) have been implicated in attention, working memory, and cognitive control (*Ociepka et al., 2024*). Alpha waves (8–12 Hz) are prominent during relaxation and are thought to inhibit irrelevant brain regions, facilitating focused attention (*Mahjoory et al., 2019*). Beta oscillations (12–30 Hz) are associated with alertness and active thinking, while gamma oscillations (30–100 Hz) are involved in higher cognitive functions such as sensory processing and perception. Previous research has shown that exercise can modulate brain activity across various frequency bands (*Fang et al., 2022*). Given the relevance of these frequency bands to cognitive function and exercise-induced brain changes, we have selected delta, theta, alpha, beta, and gamma frequencies for analysis in our study. In addition to microstates, EEG complexity, which evaluates the degree of coordination between different brain regions based on spatial distribution, is a new research hotspot in neuroscience. EEG complexity reflects spatial architecture of the dynamic functional network of the brain (*Sun et al., 2020*). Research has shown that abnormal changes in EEG complexity are closely related to various diseases such as schizophrenia (*Zhang et al., 2021*), depression (*Čukić et al., 2020*), and Alzheimer's disease (*Yang et al., 2013*).

Traditional EEG measurement methods based on chaos or nonlinear system theory in spectrum analysis, such as the Intrinsic Fuzzy Entropy method (*Valencia et al., 2019*) and Lempel Ziv Complexity (*Borowska, 2021*), can only describe the complexity of single-channel time series, and cannot characterize the coordinated changes reflecting brain functional status in the whole brain or specific regions. Recently, as a non-referenced measurement method, omega complexity has attracted wide attention from neuroscience researchers (*Gao et al., 2017*). Omega complexity is calculated using Fourier transform, which computes the entropy of the λ spectrum of the EEG data cross-spectrum matrix at each frequency point, quantifying the characteristic of dynamic synchronization between different brain areas/channels, including multi-channel, full-time, whole-brain, or local representation of the spatial features of brain functional status. Previous research has shown that sleep deprivation can cause a decrease in β−2 (20–30 Hz) and γ (30–45 Hz)-wave full-brain omega complexity in a subclinical depression group, and complexity characteristics can be used as early diagnostic indicators for depression (*Zhao et al., 2022*). Given the relevance of these frequency bands to cognitive function and exercise-induced brain changes, we have selected delta, theta, alpha, beta, and gamma frequencies for analysis in our study.

Dragon boat racing is a traditional water sport that originated from the dragon culture of China over 2,000 years ago and has now become a popular international water sports event (*Ho et al., 2013*). A dragon boat race involves 20 paddlers, a drummer, and a steersperson who propel a long canoe-like boat over a race distance of 200 to 2,000 m. Studies have shown that dragon boating has a positive impact on the physical, functional, psychological, and social challenges faced by breast cancer survivors, including enhancing meaning and purpose, self-confidence, self-esteem, sense of control, and social interaction (*Iacorossi et al., 2019*; *van Someren & Palmer, 2003*). Dragon boating involves both aerobic and anaerobic exercise. *Ho et al. (2013)* conducted simulated 200 and 500 m races with 11 male national dragon boat athletes in Japan, and the results showed that the aerobic energy supply of these athletes surpassed 50%. Other research has shown that anaerobic capacity is a more sensitive indicator for evaluating elite athletes during paddling sprinting (*van Someren & Palmer, 2003*). The 1,000-m race is a widely recognized and frequently used distance in dragon-boat competitions, providing a well-established benchmark for assessing performance and physiological responses. This distance strikes a balance between capturing the high-intensity aspects of the sport and allowing participants to sustain the effort for a duration that is relevant to our study objectives. While other distances, such as 200, 500, or 2,000 m, are also used in dragon-boat racing, the 1,000-m distance was deemed suitable for our investigation due to its prevalence in competitions and its representation of a challenging yet manageable duration for acute exercise effects.

Dragon-boat racing is characterized by dynamic nature, involving high-intensity paddling bursts interspersed with brief recovery periods. This dynamic and demanding form of exercise combines elements of both aerobic and anaerobic activities. Unlike traditional aerobic exercises, which typically involve sustained moderate-intensity activities, dragon-boat racing requires rapid bursts of power and coordination, akin to anaerobic exercises. However, the prolonged duration of dragon-boat races also places

demands on aerobic endurance, distinguishing it from purely anaerobic activities. Research on high-intensity interval training (HIIT) has demonstrated its beneficial effects on cardiovascular fitness, metabolic health, and cognitive function (*Martland et al., 2020*; *Mekari et al., 2020*). Despite this, the neural mechanisms underlying these effects, particularly in the context of acute exercise, remain less understood (*Wu et al., 2022*). Drawing parallels between dragon-boat training and HIIT allows us to leverage existing HIIT literature to inform our interpretation of the neural changes observed following dragon-boat exercise. Additionally, dragon-boat training involves coordinated movements of the upper body, core stability, and cardiovascular conditioning. This activity requires synchronized paddling in response to auditory cues, promoting both physical and cognitive engagement. Dragon-boat training has been associated with various cognitive and psychological benefits, including improved attention, executive function, and mood (*Wu et al., 2023*; *Xu et al., 2021*). Investigating its effects on neural activity is therefore of particular interest for understanding the underlying mechanisms of these cognitive enhancements. In terms of neural markers, EEG-derived measures, such as microstate analysis and complexity metrics, were selected due to their sensitivity to temporal dynamics and global connectivity patterns of brain activity (*Gao et al., 2017*; *Khanna et al., 2015*). Microstate analysis provides insights into the functional organization of the brain by identifying stable states of neural activity, while complexity metrics offer a quantitative assessment of the brain's information processing capacity. By examining changes in these neural markers following dragon-boat training, we aim to elucidate how specific physical activities impact brain function at both the macroscopic and microscopic levels.

This study aims to investigate the impact of dragon-boat training on brain function using EEG. To explore the reshaping effect of exercise at multiscale (spatial and temporal) levels on large-scale brain networks, this study recruited professional dragon boat athletes and ordinary university students and conducted a 1,000-m paddling test. The subjects' EEG data were collected before and after exercise, and changes in microstates and brain spatial complexity were observed. We hypothesize that participation in dragon-boat training will result in specific changes in neural activity, cognitive function, and related outcomes compared to more traditional exercise modalities. Specifically, we anticipate that individuals engaging in dragon-boat training will exhibit enhanced neural synchronization, improved cognitive performance, and greater functional connectivity within brain networks associated with attention and motor control. These effects are expected to be evident both acutely following a single training session and longitudinally over time with regular training participation.

## METHODS

### Participants

A power analysis was conducted to determine the minimum detectable effect for a repeated measure, within-between interaction, ANOVA sample size calculation using G*Power 3.1 (*Faul et al., 2007*). The study utilized values of α = 0.05, power = 0.80, effect
**Table 1 Demographic information of participants (mean ± SD).**

| Index | Results | |
|---|---|---|
| | **Amateur group ($N = 23$)** | **Professional group ($N = 24$)** |
| Age (years) | 20.23 ± 1.59 | 20.36 ± 1.01 |
| Height (cm) | 172.87 ± 4.13 | 183.36 ± 5.75 |
| Weight (kg) | 72.80 ± 9.84 | 84.50 ± 9.53 |
| Training years (years) | 0.71 ± 0.36 | 4.57 ± 1.23 |
| Simulation test results (s) | 434.86 ± 37.79 | 344.78 ± 30.80 |

size f = 0.25 (corresponding to $*\eta^2 = 0.5$), with two groups (professional and amateur groups) and four measurements (pre-post measurements × microstate type) in *a priori* type of power analysis. The effect size of f = 0.25 was chosen based on previous studies and meta-analyses that reported medium effect sizes for the acute effects of exercise on EEG and cognitive functions (*Chang et al., 2012*; *Smith et al., 2010*). The minimum sample size required was determined to be $N = 36$. A total of 50 right-handed male participants were recruited for the study, including 25 professional and 25 amateur dragon boat athletes, from Shanghai Maritime University. All the professional dragon boat athletes were high-level athletes who participated in the National University Dragon Boat Championships: six at National Level 1 and 18 at National Level 2. The basic information on the subjects is summarized in Table 1. All participants were healthy with no history of mental illness or major injuries. Study approval was obtained from the Ethics Board of Shaanxi Normal University (No. 202116012) in accordance with the Declaration of Helsinki. All subjects signed an informed consent form on site after understanding the procedure and purpose of this experiment.

## Experimental equipment and procedures

This study used a comparative experimental design to investigate the impact of dragon boat training on brain function in both professional and amateur groups. The flow chart of the experiment is illustrated in Fig. 1.

Participants underwent resting EEG signal collection on two separate occasions: once 30 min before and once 15 min after a single dragon boat training session. After the initial EEG data collection, participants performed a 10-min warm-up, followed by a 1,000-m dragon boat simulation training session. This distance is common in dragon boat competitions, serving as a well-established benchmark for assessing both performance and physiological responses for dragon boat athletes. The D1-M dragon boat dynamometer (KayakPro, Miami, FL, USA) was used to simulate dragon boat training, offering a range of resistance levels from 0 to 9. To replicate the dragon boat race experience, the 1,000-m test distance was completed using the highest wind resistance level (9th gear) as the exercise load. After the training session, EEG data were collected again after a 15-min rest period.

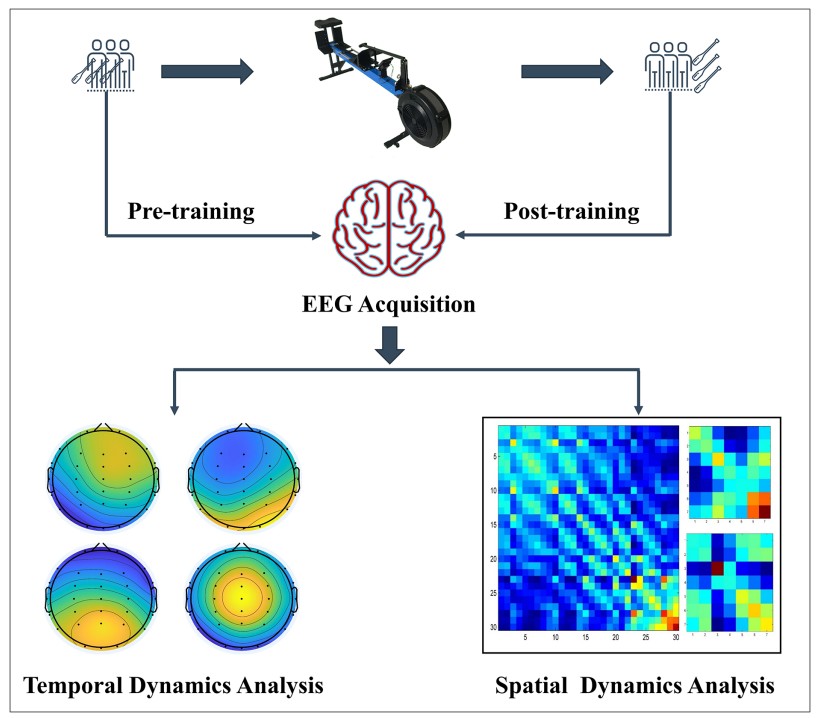

**Figure 1 Experimental flow chart.**

## EEG acquisition and preprocessing

The EEG data collection took place between 9:00–11:00 in the morning in a quiet, noise-free, indoor setting. The EEG data was collected using the Brain Vision Recorder and Neuroscan EEG recorder, with 32 EEG caps placed according to the international 10–20 standard system. Thirty-two channels were recorded using a band-pass filter of 0.1–100 Hz and continuous sampling at 1,024 Hz. M1 and M2 of the left and right mastoid processes were used as references, with electrode scalp resistance minimized to less than 10 kΩ. Before EEG data collection, participants were instructed to clean their heads and avoid using hair attachments. During EEG recording, participants sat comfortably in a dimly-lit, noiseless, and electrically-shielded room, with eyes closed to minimize artifacts.

The EEG data was preprocessed using MATLAB's open-source toolkit, EEGLAB (Version R2013b; EEGLAB, San Diego, CA, USA). A sampling rate of 500 Hz was applied, and band-pass and notch filters were used to eliminate artifacts. The data was segmented into 2,000-millisecond intervals, and independent component analysis was used to remove non-physiological artifacts. EEG amplitude values exceeding ±100 μV were removed, and a common average reference value was calculated post-artifact removal. Finally, the clean EEG data was saved for further processing.

## EEG microstate analysis

In this study, the microstate analysis followed the methods of previous studies by performing digital bandpass filtering from two to 20 Hz on the processed EEG data
(*Lehmann et al., 2005*; *Schlegel et al., 2012*). For the microstate analysis, the multichannel EEG signal was considered a series of instantaneous topographies of electric potentials. The maximum signal-to-noise ratio points were identified by calculating the global field power (GFP) for each terrain in the time series. The GFP at each time point was equivalent to the root mean squared on electrodes:

$$GFP(t) = \sqrt{\frac{\sum_{i=1}^{n} (v_i(t) - \bar{v}(t))^2}{n}}$$

where $v_i(t)$ is the voltage at electrode $i$ at time $t$, $\bar{v}(t)$ is the mean voltage across all electrodes at time $t$, and $n$ is the number of electrodes.

The EEG was further analyzed at the GFP peak. The microstate analyses were based on the Topographic Atomize and Agglomerate Hierarchical Clustering (T-AAHC) algorithm (*Brunet, Murray & Michel, 2011*). The T-AAHC algorithm was used for topology identification and cross-validation was used to determine the optimal number of clusters. In the EEG microstate analysis, parameters were set in the EEGLAB tool, as follows: the minimum number of classes was set to four, the maximum number of classes was set to six, with 10 restarts and a maximum of 100 maps used. For the label smoothing window and non-smoothness penalty, a window of 150 milliseconds and a non-smoothness penalty of 0.2 were applied. Global explanation variance (GEV) was used to judge the fitting effect, with a higher value indicating a better fit. The spatial correlation between cluster-level microstate classification topographies was used to determine the sequence of microstate classification. Finally, each original map was allocated to an EEG microstate using the maximum spatial correlation coefficient between the original map tested and the group-level microstate map.

## Omega complexity analysis

The omega complexity of EEG was calculated using the method proposed by *Gao et al. (2017)*, which involves transforming 2-s segments from all electrodes to the frequency domain using Fourier transforms and computing the cross-spectral matrices for the 30 channels. The resulting omega complexity ranges from one to K and reflects the spatial complexity of a given set of EEGs (*Gao et al., 2017*; *Kondakor et al., 2005*). A smaller omega value indicates high synchronization with a single mode, while a larger value indicates poor synchronization with multiple modes.

$$C = \frac{1}{N} \sum_{i=1}^{N} u_i * u_i^T$$

where $K$ is the number of electrodes and $N$ is the EEG signal length.

Then the eigenvalue $\lambda_1, \ldots, \lambda_k$ of the covariance matrix C was calculated. Next, the normalized feature $\lambda_i'$ was calculated, as follows:

$$\lambda_i' = \frac{\lambda_i}{\sum \lambda_i}$$

The definition of omega was defined as follows:

$$\log(\Omega) = -\sum \lambda_i' * \log(\lambda_i')$$

Using that definition of omega, the omega ($\Omega$) spatial complexity was calculated, as follows:

To calculate the omega complexity for each subject, the values for each frequency point were averaged over epochs, and the complexity of the $\delta$ (1–4 Hz), $\theta$ (4–7 Hz), $\alpha-1$ (8–10 Hz), $\alpha-2$ (10–13 Hz), $\beta$ (13–30 Hz), and $\gamma$ (30–45 Hz) bands was calculated. Each frequency band's global complexity was also determined.

## Statistical analysis

All the statistical analyses were performed using SPSS (23.0; SPSS, Inc., Chicago, IL, United States). The Wilcoxon test was used for non-normally distributed data. A three-way repeated-measures analysis of variance (ANOVA) with two groups (amateur group and professional group) × two times (pre-test and post-test) × four microstates (A, B, C, D) was performed for the duration, occurrence, contribution, and transition probability of each microstate. Separate two-way ANOVA with two groups (professional and amateur groups) × two times (pre-test and post-test) was also performed for the omega complexities in each frequency band. The level of significance was set at $p < 0.05$. The Bonferroni-corrected method was performed for *post hoc* testing of significant main effects to minimize the risk of type I error. Effect size in all ANOVA analyses was reported by partial eta squared ($\eta^2$), where 0.05 represents a small effect, 0.10 represents a medium effect, and 0.20 represents a large effect (*Faul et al., 2009*).

## RESULTS

### EEG microstate

Due to physical limitations, three participants were not included in the final experiment, making the final study population 47, including 23 amateurs and 24 professional dragon boaters. To compare the characteristics of the fitted microstate sequences for each group, the proportion of four representative microstates was calculated before and after a single dragon boat exercise. EEG data from both groups of participants were analyzed to determine any differences in their microstate patterns. The results are shown in Fig. 2. Before the dragon boat exercise, there was no significant difference in GEV between the amateur group and the professional group (73.039% ± 0.047 *vs.* 73.249% ± 0.055, $p = 0.575$). There was also no significant difference in GEV value between the two groups after the single dragon boat exercise (71.245% ± 0.071 *vs.* 72.432% ± 0.062, $p = 0.625$).

### Comparison of EEG microstate time parameters

The duration, occurrence, and coverage of microstate results are shown in Table 2.

The duration results showed that the main effect of group (amateur *vs.* professional) was not significant ($F_{(1, 90)} = 0.140$, $p = 0.709$, $\eta^2 = 0.002$); the main effect of time was not significant ($F_{(1, 90)} = 0.214$, $p = 0.645$, $\eta^2 = 0.002$); but the main effect of microstates class

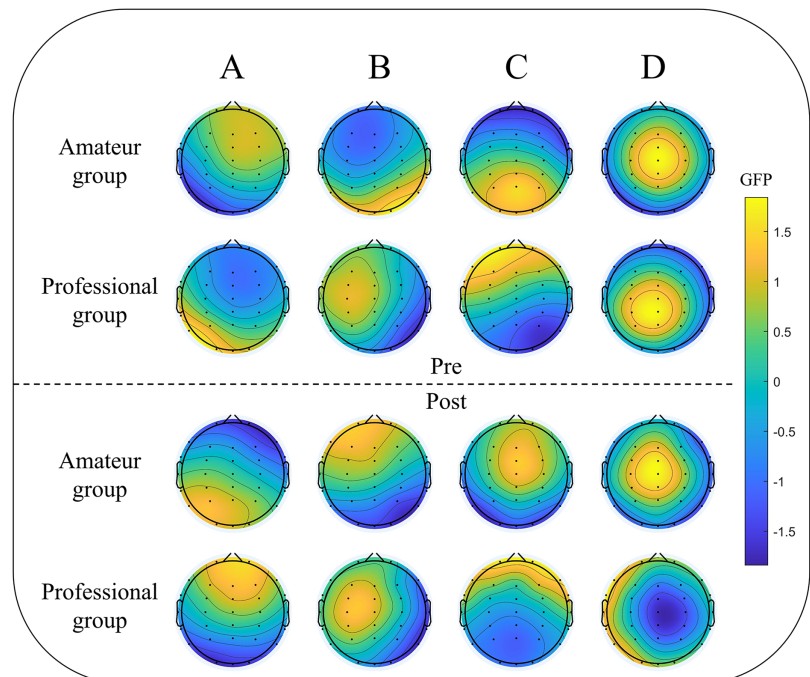

**Figure 2 Group level mapping of EEG microstates in two groups of participants pre- and post-dragon boat exercise.** (A–D) represent the four clustering states of microstates (A–D), respectively; pre: pre-dragon boat exercise; post: post-dragon boat exercise. The color scheme was determined based on the global field power (GFP) values.

was significant (F(3, 88) = 3.711, $p$ = 0.014, $\eta^2$ = 0.112). *Post hoc* analysis showed that the duration of microstate D was significantly higher for the professional group than the amateur group before training. In the amateur group, the duration of microstate C was significantly longer after exercise than it was before exercise.

The occurrence results indicated that the main effect of group was not significant (F(1, 90) = 3.197, $p$ = 0.077, $\eta^2$ = 0.034); the main effect of time was not significant (F(1, 90) = 0.049, $p$ = 0.826, $\eta^2$ = 0.001); the main effect of microstates class was not significant (F(1, 90) = 2.963, $p$ = 1.419, $\eta^2$ = 0.016); but there was a significant interaction between microstates class and group (F(3, 90) = 3.705, $p$ = 0.028, $\eta^2$ = 0.003). A simple effect analysis showed that after exercise, the occurrence of microstate D in the professional group was significantly higher than in the amateur group.

The contribution results showed that the main effect of group was not significant (F(1, 90) = 0.303, $p$ = 0.584, $\eta^2$ = 0.003); the main effect of time was not significant (F(1, 47) = 0.351, $p$ = 0.555, $\eta^2$ = 0.004); the main effect of microstates class was not significant (F(3, 88) = 2.972, $p$ = 0.327, $\eta^2$ = 0.038); but there was a significant interaction effect between microstate type and group (F(3, 88) = 3.930, $p$ = 0.009, $\eta^2$ = 0.042). A simple effect analysis showed that the contribution of microstate D in the professional group was significantly higher than in the amateur group both before and after exercise.

**Table 2 Microstate results before and after single simulation training for professional and amateur groups.**

| Time parameters | Pre | | Post | | Cohen's d Amateur *vs.* professional | |
|---|---|---|---|---|---|---|
| | Amateur | Professional | Amateur | Professional | Before | After |
| Total time (s) | 294.843 ± 47.525 | 322.467 ± 26.599 | 283.779 ± 56.069 | 303.613 ± 37.332 | −0.717 | −0.416 |
| Duration (s) | | | | | | |
| A | 0.073 ± 0.018 | 0.072 ± 0.010 | 0.074 ± 0.021 | 0.070 ± 0.011 | 0.068 | 0.239 |
| B | 0.065 ± 0.019 | 0.068 ± 0.022 | 0.061 ± 0.008 | 0.066 ± 0.012 | −0.146 | −0.490 |
| C | 0.063 ± 0.012 | 0.064 ± 0.015 | 0.071 ± 0.017[*] | 0.065 ± 0.012 | −0.074 | 0.408 |
| D | 0.064 ± 0.016 | 0.073 ± 0.013[#] | 0.070 ± 0.020 | 0.069 ± 0.032 | −0.617 | 0.037 |
| Mean duration | 0.071 ± 0.007 | 0.073 ± 0.011 | 0.072 ± 0.006 | 0.069 ± 0.009 | −0.217 | 0.392 |
| Occurrence | | | | | | |
| A | 3.982 ± 1.212 | 3.384 ± 1.523 | 3.900 ± 1.321 | 3.297 ± 1.139 | 0.434 | 0.489 |
| B | 3.556 ± 1.470 | 3.920 ± 1.508 | 3.267 ± 1.062 | 3.428 ± 1.128 | −0.244 | −0.147 |
| C | 3.281 ± 1.245 | 3.875 ± 1.280 | 3.482 ± 1.474 | 3.955 ± 1.081 | −0.470 | −0.366 |
| D | 3.767 ± 1.306 | 3.695 ± 1.713 | 3.653 ± 0.992 | 4.326 ± 0.835[#] | 0.047 | −0.734 |
| Mean occurrence | 14.572 ± 1.231 | 14.784 ± 1.634 | 14.429 ± 1.088 | 15.007 ± 1.486 | −0.147 | −0.444 |
| Contribution (%) | | | | | | |
| A | 0.278 ± 0.138 | 0.259 ± 0.136 | 0.243 ± 0.153 | 0.248 ± 0.084 | 0.139 | −0.041 |
| B | 0.230 ± 0.138 | 0.279 ± 0.165 | 0.206 ± 0.086 | 0.218 ± 0.010 | −0.322 | −0.196 |
| C | 0.204 ± 0.100 | 0.250 ± 0.132 | 0.237 ± 0.133 | 0.276 ± 0.128 | −0.393 | −0.299 |
| D | 0.268 ± 0.148 | 0.320 ± 0.123[#] | 0.250 ± 0.089 | 0.292 ± 0.154[#] | −0.382 | −0.333 |

**Note:**
[#]Paired samples t-test; [*]Independent samples t-test; [#] and [*] indicate: $p < 0.05$ for a statistical difference.

## Comparison of microstate transition probability

Since the microstate transition probability data was not normally distributed, the Mann-Whitney U test was used to compare the transition probabilities of various groups, while the Wilcoxon signed-rank test was used to compare the transition probabilities of the same group before and after exercise. The statistical results are presented in the Supplemental File. The transitions of the four types of microstates are illustrated in Fig. 3.

Before the exercise, the professional group had a higher probability of converting from microstate A to microstate D ($Z = −2.42$, $p = 0.01$) than the amateur group. After exercise, the professional group had a significantly higher probability of converting from C to D than the control group ($Z = −2.43$, $p = 0.01$). In the amateur group, after exercise, the transition probabilities from A to B ($Z = −2.159$, $p = 0.03$) were significantly lower and the transition probabilities from B to C ($Z = −2.19$, $p = 0.03$) were significantly higher than before the exercise. In the professional group, compared to before exercise, the transition probability from microstate D to C ($Z = −2.029$, $p = 0.043$) in the professional group was significantly reduced and the transition probability from B to C was significantly increased ($Z = −2.229$, $p = 0.026$) after exercise.

## Omega complexity

Omega complexity results are shown in Table 3.

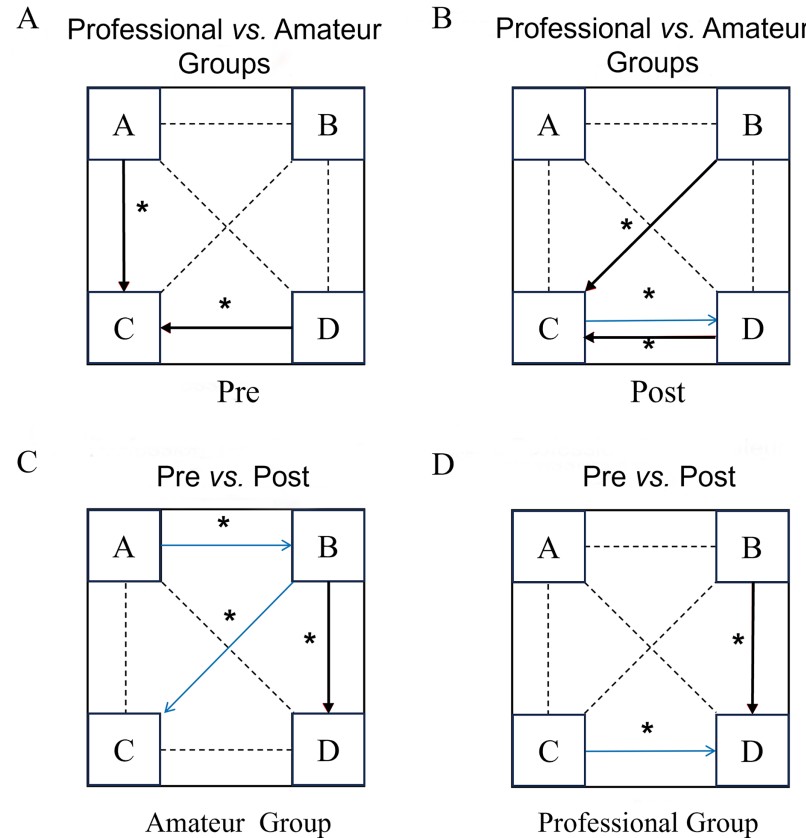

**Figure 3 Comparison of microstate transition results before and after the dragon boat exercise in the professional and amateur groups.** (A–D) The four clustering states of microstates (A–D) respectively; pre: pre-dragon boat exercise; post: post-dragon boat exercise. Red arrow: increase, blue arrow: decrease. An asterisk (*) indicates: $p < 0.05$ for a statistical difference.

**Table 3 Comparison of the differences in global spatial complexity between the two groups before and after the dragon boat training session.**

| Frequency band | Pre | | Post | | Cohen's d AG *vs.* PG | |
|---|---|---|---|---|---|---|
| | Amateur group | Professional group | Amateur group | Professional group | Pre | Post |
| Delta | 4.038 ± 0.530 | 3.946 ± 0.755 | 4.265 ± 0.637 | 3.991 ± 0.467 | 0.141 | 0.491 |
| Theta | 4.130 ± 0.528 | 4.095 ± 0.750 | 4.343 ± 0.635 | 4.096 ± 0.456 | 0.054 | 0.447 |
| Alpha-1 | 4.241 ± 0.534 | 3.875 ± 0.567* | 4.431 ± 0.660 | 4.087 ± 0.401* | 0.665 | 0.630 |
| Alpha-2 | 4.146 ± 0.522 | 4.012 ± 0.750 | 4.529 ± 0.563[#] | 4.163 ± 0.469* | 0.207 | 0.706 |
| Beta-1 | 4.122 ± 0.521 | 4.011 ± 0.724 | 4.343 ± 0.630 | 4.100 ± 0.458 | 0.176 | 0.441 |
| Beta-2 | 4.087 ± 0.520 | 3.967 ± 0.715 | 4.320 ± 0.577[#] | 4.053 ± 0.440* | 0.192 | 0.520 |
| Gamma-1 | 4.095 ± 0.523 | 3.981 ± 0.711 | 4.332 ± 0.619 | 4.067 ± 0.439 | 0.183 | 0.494 |

Notes:
Pre: before dragon boat exercise; Post: after dragon boat exercise. [#]Paired samples t-test; *Independent samples t-test; [#] and * indicate: $p < 0.05$ for a statistical difference.
AG, Amateur Group; PG, Professional Group.

$\alpha-1$ (8–10 Hz): The main effect of group was significant (F(1, 90) = 9.872, $p$ = 0.002, $\eta^2$ = 0.099), with *post-hoc* analysis revealing that the omega complexity of the professional group was significantly lower than that of the control group before and after the intervention, with $p$ values of 0.024 and 0.034, respectively. The main effect of time was not significant (F(1, 90) = 3.173, $p$ = 0.078, $\eta^2$ = 0.034), and the interaction effect between time and group was not significant (F(1, 90) = 0.009, $p$ = 0.923, $\eta^2$ < 0.000).

$\alpha-2$ (10–13 Hz): The main effect of group was significant (F(1, 90) = 4.800, $p$ = 0.031, $\eta^2$ = 0.041), with *post-hoc* analysis showing that the omega complexity of the professional group was significantly lower than that of the control group after the intervention ($p$ = 0.026). The main effect of time was significant (F(1, 90) = 5.490, $p$ = 0.021, $\eta^2$ = 0.057) and further revealed that the omega intensity of the amateur (control) group was significantly higher after the intervention than before ($p$ = 0.021).

$\beta$ (13–30 Hz): The main effect of group was significant (F(1, 90) = 4.398, $p$ = 0.039, $\eta^2$ = 0.047), with *post-hoc* analysis showing that the omega complexity of the professional group was significantly lower than that of the control group after the intervention ($p$ = 0.027), and the complexity of the amateur group was significantly higher after the intervention than before ($p$ = 0.046). The interaction between group and time was not significant (F(1, 90) = 1.182, $p$ = 0.280, $\eta^2$ = 0.013).

$\delta$ (1–4 Hz): The main effects of group (F(1, 90) = 2.148, $p$ = 0.146, $\eta^2$ = 0.023), time (F(1, 90) = 1.171, $p$ = 0.282, $\eta^2$ = 0.013), and the interaction between group and time (F(1, 90) = 0.522, $p$ = 0.472, $\eta^2$ = 0.006) were not significant.

$\theta$ (4–7 Hz): The main effects of group (F(1, 90) = 1.270, $p$ = 0.263, $\eta^2$ = 0.014), time (F(1, 90) = 0.737, $p$ = 0.393, $\eta^2$ = 0.008), and the interaction between group and time (F(1, 90) = 0.733, $p$ = 0.394, $\eta^2$ = 0.008) were not significant.

$\gamma$ (30–45 Hz): The main effects of group (F(1, 90) = 1.817, $p$ = 0.181, $\eta^2$ = 0.020), time (F(1, 90) = 2.492, $p$ = 0.118, $\eta^2$ = 0.027), and the interaction between group and time (F(1, 90) = 0.397, $p$ = 0.530, $\eta^2$ = 0.004) were not significant.

## DISCUSSION

This study explores the effects of acute dragon boat exercise on the spatiotemporal dynamics of brain neural activity using EEG. The findings suggest that a singular dragon boat exercise session significantly improves the functional network of the brain and enhances its dynamic coordination across diverse temporal and spatial scales. Moreover, this heightened brain synchronization persisted at elevated levels even following intense, full-strength paddling.

Resting-state EEG microstates have become a biological marker for diagnosing various diseases (*Khanna et al., 2015*). However, despite studies that have applied microstates to the evaluation of cognitive load and fatigue (*Guan et al., 2022*), their application in the field of exercise is in its infancy. Microstate D has been shown to be associated with attention-related (*Custo et al., 2017*). Previous studies have identified microstate D as a frontal area network that maintains attention and cognitive control (*Brechet et al., 2019*; *Custo et al., 2017*). The duration of microstate D is assumed to reflect the activation of the attentional system, and its variability should be considered together with the co-regulation

of microstate C (*Custo et al., 2017*). The results of this study showed that the duration and coverage of microstate D in the professional group were both significantly higher than in the amateur group before exercise, which is consistent with the conclusion of (*Gu et al., 2022*). *Gu et al. (2022)* observed changes in the brain EEG of elite and expert archers during resting state and the aiming period and found that the duration, frequency, and coverage of microstate D were all higher in the expert group.

Many underlying mechanisms may be involved in the effects of dragon boat training on EEG microstates. One possible mechanism at play is neural plasticity (*Dayan & Cohen, 2011*). The influence of dragon boat training on microstate D may be related to permanent changes in the brain synaptic events caused by continuous learning and motor skill memory. The activating effect of the attention network may also help explain the differences in microstate D observed among different groups of participants undergoing dragon boat training. Recent research has indicated that the regional activation of microstate D is associated with the frontal attention network (*Spring, Tomescu & Barral, 2017*), which is a functional system activated by attention tasks (*Fumoto et al., 2010*). Recent studies have shown that traditional Chinese mind-body exercise (Baduanjin) and brisk walking can activate the dorsal attention network of the brain and improve the level of attention in patients with mild cognitive impairment (*Xia et al., 2019*). This evidence suggests that dragon boat exercise may activate and continuously strengthen the activity of the dorsal attention network in the brain, which is constantly reinforced by relevant cognitive tasks.

The regional activation of microstate C, primarily in the anterior insula/frontal operculum and anterior cingulate cortex, is closely associated with the salience network, which plays a crucial role in the brain's ability to switch between the central executive and default mode networks (*Sridharan, Levitin & Menon, 2008*). It was hypothesized that acute dragon boat training affects microstate C because of its co-regulatory role with microstate D in attention maintenance and cognitive control (*Seitzman et al., 2017*). Consistent with this hypothesis, the duration of microstate C in the amateur group significantly increased after exercise. This finding is in line with the findings of *Spring, Tomescu & Barral (2017)* and their subsequent observations: the temporal parameters of microstate C did not recover to baseline during a 60-min passive recovery period, and the reorganization of temporal parameters of the microstate C after exercise may reflect the activation status of the anterior insula and anterior cingulate cortex (*Spring, Bourdillon & Barral, 2018*). The salience network consists of two key nodes, the anterior insula and the anterior cingulate cortex, which receive and interpret somatosensory input from the cardiovascular (autonomic) and muscular systems, respectively (*Menon, 2015*). Research has shown that signals from the muscle (*via* the spinal cortical pathway (*Enders & Nigg, 2016*; *Yang et al., 2010*)) and cardiovascular system (*via* vagal nerve input; (*Benarroch, 2020*)) during physical exercise gain prominence in sustained body sensation and autonomic information flow. The changes in microstate C induced by dragon boat training, as observed in the present study, may involve the activation of the ipsilateral sensorimotor area and cingulate gyrus and alterations in autonomic function or activation of the salience network through the anterior insula. These changes could be the results of changes in autonomic function or

of peripheral muscle fatigue (*Spring, Tomescu & Barral, 2017*) and its effect on activation in these brain regions. Furthermore, while acute exercise can effectively activate the salience network, the effect of an exercise intervention on microstate C may depend on the type of exercise. A study examining spontaneous EEG activity after a mountain ultra-marathon observed a significant decrease in the mean duration and time coverage of microstate C, indicating that overtraining may lead to desensitization of the salience network's negative feedback loop (muscle metabolic accumulation, inflammation information; (*Spring et al., 2022*). These changes to microstate C were the opposite of those observed in the present study.

The present study found that before exercise, the transition from microstate A to microstate D in the professional group was significantly higher than in the amateur group. This may be an underlying factor for the increase in the duration and coverage of microstate D observed in the professional dragon boat group before exercise. After exercise, the transition probabilities from microstate B to D and from C to D were significantly higher in the professional group than in the amateur group, while the transition probability from D to C was significantly lower in the professional group than in the amateur group. These results suggest that the professional group had a smaller decrease in attention level after the dragon boat training, which may because the long-term training of the professional athletes enhanced the baseline state of attention networks in the brain. In the amateur group after exercise, the transition probabilities from microstate A to B and from B to D were significantly lower than before exercise, while the transition probability from microstate B to microstate C was significantly higher than before exercise. These results indicate that acute high-intensity dragon boat training promotes the transfer of other networks to the salience network and the dorsal attention network in response to the exercise training. *Spring (2018)* reported a correlation between peripheral muscle fatigue (which can cause a decrease in cortical muscle coupling) and an increase in the average duration of microstate C during a single acute training task.

*Spring (2018)* also found that the communication of the autonomic nervous system induced by exercise was considered a potential regulator of spontaneous heart rate adjustment and static EEG microstate. The effects of dragon boat exercise on various time parameters and transition probabilities of microstates observed in this study may be related to muscle or autonomic nervous system changes. However, more research is needed to elucidate other mechanisms by which exercise-induced changes in microstates occur.

The measurement of brain entropy indicates the degree of orderliness in the brain, which can reveal the level of synchronization and temporal changes across different brain regions in response to stimuli. As such, brain entropy has been recognized as a valuable biological marker for diagnosing neurological disorders (*Jacob & Nair, 2019*).

Complexity has also become a hot topic in exercise science research (*Wang et al., 2019*; *Zhu, 2021*) and is often employed as an important reference index for elucidating the central nervous system mechanisms underlying exercise performance. Research has shown that exercise training can effectively reduce the complexity level of the brain and complexity is negatively correlated with exercise performance. Studies

by *Babiloni et al. (2010b)* and *Del Percio et al. (2010)* have shown that professional athletes have more concentrated information in their brains, greater orderliness, and higher anti-interference ability. *Babiloni et al. (2010a)* conducted a series of studies analyzing scalp EEG activity in professional athletes, amateur athletes, and ordinary people to compare the efficiency of the underlying neural synchronization mechanism. They found that the basal-cortical neural synchronization efficiency of professional athletes during closed-eye rest was enhanced in the alpha rhythm of the brain's electrical activity, suggesting that the brains of professional athletes have higher neural efficiency. The findings of the present study (Table 3) show that before exercise, the professional group had lower complexity in all seven frequency bands compared to the control group, and the difference in the alpha-1 band was significant. These results suggest that the brains of elite athletes have higher neural efficiency. This may be due to two possible mechanisms: (1) long-term dragon boat training may cause persistent changes in the neural discharge rate, synapses, and brain topography of the primary motor cortex, promoting greater coordination, stability, and order in different functional brain networks as well as enhanced synergistic effects between networks; (2) long-term dragon boat training may also be related to increased brain metabolism and enhanced cardiovascular function, which may increase oscillatory changes in brain electrical activity, attention, and alertness, as well as enhance cortical and performance efficiency (*Hogan et al., 2015*). Notably, compared to the amateur (control) group, the professional group showed significantly lower complexity in the alpha-2 and beta-2 bands, suggesting that long-term dragon boat training can effectively reduce the complexity induced by high-intensity exercise (which may be related to exercise fatigue), possibly because of an improvement in the efficiency of neural synchronization in the underlying background of basic brain electrical activity.

Previous research on the relationship between exercise and complexity has primarily focused on whole-brain complexity and the alpha band frequency (*Pedroso et al., 2021*). This study is the first to observe changes in complexity in different frequency bands (seven bands) and brain regions (frontal and posterior). The results showed that before exercise, the professional group had significantly lower alpha-1 and beta-2 complexity in the frontal region compared to the control group; after exercise, alpha-1, alpha-2, and beta-1 complexity in the frontal region were significantly lower in the professional group than in the control group. Notably, both before and after exercise, the differences in complexity between the two groups were mainly concentrated in the alpha and beta frequency bands, and all of the different regions were concentrated in the anterior region of the brain, suggesting that the long-term impact of dragon boat training on the brain mainly focuses on the alpha and beta frequency bands in the frontal (prefrontal and temporal) regions of the brain. Recent research has indicated that acute exercise can increase the excitability of the motor cortex and enhance the motor learning ability of healthy individuals, known as the exercise start-up effect (*Li et al., 2019*). Performing oddball paradigms and transient oscillatory brain activity tasks during moderate to high-intensity aerobic training can increase power across the entire spectrum (*Ciria et al., 2019*). Given the negative correlation between the power spectrum and complexity, this study provides new evidence

from the perspective of complexity changes to elucidate the mechanisms underlying the exercise start-up effect.

While this study provides valuable insights into immediate changes in neural activity and cognition, it does not capture the potential long-term adaptations that may result from prolonged participation in dragon boat racing. To address this limitation and further advance the field, future longitudinal studies that delve into the chronic effects of dragon boat training are recommended. Longitudinal research could explore how sustained engagement in this unique exercise modality impacts neural plasticity, cognitive function, and other relevant outcomes over an extended period. Such studies would contribute to a more comprehensive understanding of the cumulative effects of dragon boat training on brain health and function. Additionally, investigating potential moderating factors, such as individual differences in fitness levels, training intensity, and duration of engagement in dragon boat racing, could enhance our understanding of the variability in responses to this exercise modality over time.

## CONCLUSION

A single dragon boat training session reshapes brain network function, as evidenced by an increase in the frequency and coverage of microstate D and its transition probability and a decrease in omega complexity throughout the brain. This study's findings on the reorganization effects of dragon boat training on the microstate and complexity of EEG may offer new direct evidence for the mechanisms underlying dragon boat training's ability to promote brain health. The results of this study suggest that dragon boat training can be used to enhance individual cognitive health or treat cognitive functional disorders or other brain disorders.

### Funding
The authors received no funding for this work.

### Competing Interests
The authors declare that they have no competing interests.

### Author Contributions
- Hongke Jiang conceived and designed the experiments, performed the experiments, analyzed the data, prepared figures and/or tables, authored or reviewed drafts of the article, and approved the final draft.
- Shanguang Zhao conceived and designed the experiments, performed the experiments, analyzed the data, prepared figures and/or tables, authored or reviewed drafts of the article, and approved the final draft.
- Qianqian Wu performed the experiments, prepared figures and/or tables, authored or reviewed drafts of the article, and approved the final draft.
- Yingying Cao performed the experiments, authored or reviewed drafts of the article, and approved the final draft.

- Wu Zhou performed the experiments, authored or reviewed drafts of the article, and approved the final draft.
- Youwu Gong performed the experiments, authored or reviewed drafts of the article, and approved the final draft.
- Changzhuan Shao performed the experiments, authored or reviewed drafts of the article, and approved the final draft.
- Aiping Chi conceived and designed the experiments, performed the experiments, authored or reviewed drafts of the article, and approved the final draft.

### Human Ethics

The following information was supplied relating to ethical approvals (*i.e.*, approving body and any reference numbers):

This research was obtained from the Ethics Board of Shaanxi Normal University (No. 202116012) in accordance with the Declaration of Helsinki, Committee.

### Data Availability

The data is available in the Supplemental File.

### Supplemental Information

Supplemental information for this article can be found online at http://dx.doi.org/10.7717/peerj.17623#supplemental-information.

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
