# Peer review of "Dragon boat exercise reshapes the temporal-spatial dynamics of the brain"

_PeerJ, doi:10.7717/peerj.17623_

## Round 0.1 · original submission · Major Revisions

We appreciate your submission and recognize its potential contribution to the field. However, to align with the standards of PeerJ, we kindly request the following revisions for clarity, coherence, and academic integrity:

Introduction of EEG Microstate Background: Please provide a brief overview of EEG microstates A-D, enhancing understanding for readers unfamiliar with this analysis technique. A supporting citation is necessary for this addition.

Inclusion of a Clear Hypothesis: The absence of a hypothesis in the original manuscript is notable. We recommend formulating and clearly stating a hypothesis that guides the research strategy and aids in interpreting the findings.

Discussion of Limitations and Future Research: It's essential to acknowledge the study's limitations, focusing primarily on the acute effects of dragon-boat training. Please include suggestions for future longitudinal studies to explore chronic effects.

Consistency in Citations and References: Ensure that all citations accurately match their corresponding references, maintaining scholarly standards and reinforcing the validity of the claims.

Rationale Behind Choosing Dragon Boating: Elaborate on why dragon boating was selected as the exercise modality, highlighting its unique aspects compared to traditional exercises.

Careful Interpretation of Cross-Sectional Results: In the conclusion, cautiously interpret the cross-sectional results, focusing on the acute effects observed rather than long-term changes.

We believe these revisions will significantly enhance the manuscript's clarity and scholarly rigor. Additional suggestions and detailed feedback are available in the reviewers' comments. We thank you for considering these revisions and look forward to receiving your revised manuscript.

**Language Note:** The review process has identified that the English language must be improved. PeerJ can provide language editing services - please contact us at [email protected] for pricing (be sure to provide your manuscript number and title). Alternatively, you should make your own arrangements to improve the language quality and provide details in your response letter. – PeerJ Staff

·

Basic reporting

no comment

Experimental design

The study presents valuable insights into the effects of dragon-boat training on brain function. The methodology is robust, and the results are clearly presented. The discussion effectively contextualizes the findings within the broader literature. Minor improvements in elaborating theoretical background, statistical justification, and language polishing could further enhance the manuscript's quality.

Validity of the findings

1. Omega Complexity Analysis (Page 161-167):

The method to calculate omega complexity is well-explained and based on established procedures. However, the manuscript could benefit from a more in-depth discussion of how omega complexity relates specifically to the exercise modality being studied (dragon-boat training).

2. Statistical Analysis (Page 184-195):

The statistical approach is comprehensive, employing two-way ANOVAs with appropriate factors and controls. The use of non-parametric tests for non-normally distributed data is a sound choice. However, the manuscript could enhance its analysis by addressing the potential for confounding variables that might influence the outcomes.

Additional comments

no comment

Reviewer 2 ·

Basic reporting

This is an interesting study with potential theoretical. However, I believe the authors need to clarify some
points and to provide additional information for improving our understanding of the area. Thus, I used a
major-minor approach to provide comments and suggestions that hopefully can be used to improve the
manuscript.

Experimental design

Please see my comments in the PDF file.

Validity of the findings

Please see my comments in the PDF file.

Annotated reviews are not available for download in order to protect the identity of reviewers who chose to remain anonymous.

Reviewer 3 ·

Basic reporting

The English language should be improved to ensure that an international audience can clearly understand your text. I have provided several examples in the attached document (with line numbers) but at present, it makes comprehension difficult. I would suggest a colleague proficient in English proofread your manuscript to help with word choice and clarity. Please also change the tense of the manuscript from present to past tense.

The introduction would benefit from a more thorough description of the past literature. Specifically, a few more sentences to describe the difference between the microstates A-D. A little background on what each one represents would help those less familiar with the analysis technique understand the manuscript better. Additionally, a description of what dragon-boat racing entails, and how it compares to more typical aerobic and anaerobic training styles (i.e. how is it similar/different, to help us contextualize is novel exercise modality). Finally, as this training style sounds similar to HIIT training, introduction would benefit from a review of this literature (acute high intensity aerobic exercise as well as acute HIIT exercise). How does this fit into the existing acute high-intensity aerobic/HIIT and EEG/cognition literature?

Experimental design

I have concerns with the clarity of the experimental design used in this study. The study design used in this manuscript is designed to investigate the acute effect of a single bout of dragon-boat training on two groups (pros and amateurs), however, there is much discussion about cross-sectional relationships as the “effect of long-term dragon-boat training”. As there was no chronic training intervention, the authors can not state definitively that the differences between these two groups at baseline are solely due to dragon-boat training. The manuscript needs to be reworded to focus on these two aims (1 being cross-sectional differences at baseline, and 2 being the acute effect of dragon-boat training). This is a critical distinction that is not clear in the current version of the manuscript.

In general, the methods are well-defined and easy to replicate. However, after reading the procedures, I am still somewhat unclear as to the exact study design and what happened on testing days. Figure 1 is helpful, but could use more text information to fully elucidate the procedures.

Validity of the findings

The authors have done well to identify a novel analysis and exercise modality. However, the benefit to the literature and to subjects in the discussion is lacking. Specifically, the overall impact of these results is insufficiently demonstrated. How do these changes fit in with the current literature?

Additionally, as mentioned above, the study design and the baseline comparisons are cross-sectional in nature, but the analysis and interpretation of the results appear to state that only dragon-boat racing could have caused these baseline differences (without taking into account any other possibilities). The acute exercise effect is somewhat blurred in the results, as it is difficult to delineate what is from the baseline group differences and what is from the acute exercise effect.

An overall clearer distinction needs to be made between the cross-sectional results (i.e. the effects at baseline) and the acute exercise effects (I.e. from the acute intervention done in this study). At present, it is difficult to discern whether the authors are discussing the acute effect of dragon-boat exercise or the cross-sectional associations with dragon-boat training status (i.e. cross-sectional differences between groups).

The results section would benefit from reporting the T/F values of all statistical group difference tests done in the manuscript. Further, the results are missing the reporting of several statistical tests which would help readers better interpret the results. Specifics are in the attached document.

Additional comments

I have attached a bullet point section-by-section review of the manuscript. In general, the results are interesting, novel, and have merit. However, the clarity, flow, study design, and interpretation need significant revisions to improve the readability of the manuscript. The differentiation between results looking at cross-sectional differences at baseline and effects of the acute exercise intervention need to be separated and clearly labeled. The baseline cross-sectional differences are interesting, and can partially be attributed to differences in training, but other possibilities should also be discussed (there are a lot of potential alternative explanations, which should be addressed).

Annotated reviews are not available for download in order to protect the identity of reviewers who chose to remain anonymous.

---

## Round 0.2 · Major Revisions

Thank you for the revisions submitted. Significant improvements have been made to your manuscript, which is greatly appreciated. However, upon careful consideration, our reviewers have highlighted several critical areas that require further clarification and deeper rationale to strengthen the research presented. Here are some significant points:

1. Clarity and Language:
The manuscript must be revised to enhance its readability and comprehension for an international audience. The clear and accessible language will allow the findings to resonate more effectively across diverse scientific communities. Simplifying complex terminology and ensuring consistent use of terminology throughout the text will contribute significantly to the overall clarity of the manuscript.

2. Literature Review Enhancement:
The introduction currently provides a basic research framework but lacks a detailed exploration of existing studies, particularly in delineating the distinctions between EEG microstates labeled A-D. For a more robust introduction, I recommend adding a thorough review of past literature focusing on these microstates, explaining what each state represents and their relevance in cognitive neuroscience. This will significantly aid readers who are less familiar with EEG analysis techniques in understanding the pivotal aspects of your research.

3. Contextualizing Dragon-Boat Racing:
This paper has a unique opportunity to elaborate on the specifics of dragon-boat racing. A comparative analysis with more conventional training methods, such as aerobic and anaerobic exercises, will enrich the reader's understanding. Descriptions should highlight how dragon-boat racing aligns with or differs from these methods, especially regarding physiological and cognitive demands. This comparison is crucial as it positions dragon-boat racing within the broader spectrum of sports science, providing a clear context for your empirical findings.

4. Connection with HIIT Training Literature:
Given the apparent similarities between the high-intensity nature of dragon-boat racing and High-Intensity Interval Training (HIIT), it is essential to integrate a review of relevant HIIT literature. Discussing acute high-intensity aerobic and HIIT exercises concerning EEG and cognitive function will not only frame your study within the existing body of knowledge but also highlight its novel contributions to understanding the effects of intense physical activity on neural processes. This linkage will also help draw more nuanced conclusions from the presented data.

5. Clarity of Research Rationale:
The manuscript explores the intriguing effects of specific physical activities on neural activity, but the rationale behind the choice of these activities and the related neural markers requires further clarification.

Please address these points thoroughly in your revised manuscript. We believe that attending to these comments will significantly enhance your work's robustness and scholarly contribution.

Looking forward to your revised submission.

Reviewer 2 ·

Basic reporting

I appreciate the authors' efforts. However, there are still some points that need clarification to enhance our understanding of the subject area. Therefore, I have employed a major-minor approach to provide comments and suggestions that will hopefully contribute to the improvement of the manuscript

Major concerns:

1. It is really hard to read without pointing out the line number for the reviewers to check every detail.
Please point out the line number precisely when authors reply reviewers. In addition, the PDF file is not
the same version of Tracked Changes file, so I don’t know which one is the final/correct version.

2. My questions were that” why chose 1000 meter rather than other distances, such as 200m, 500m, or
2000 meter? What is the rationale behind that? The explanations that authors replied in the response
letter were good. However, I did not see any relevant information about that in the introduction. Please
add it in the introduction.

3. The hypothesis presented lacks specificity for conducting empirical research. While the authors
anticipate that the unique characteristics of the study may result in differential effects on neural activity,
cognitive function, and other related outcomes compared to more traditional exercise modalities, the
hypothesis itself is not specific enough. It is crucial to ensure thoroughness when formulating the
research hypothesis, providing specific expectations of the results that this study aims to uncover.

4. I would like to exam the analysis code for EEG Microstate Analysis and Omega Complexity Analysis in the MATLAB (or showing EEG Microstate Analysis and Omega Complexity Analysis in the video or
specific flowchart) that authors mentioned that they have placed it in the supplementary file, but I did not see any codes, videos, or flowcharts in the supplementary file.

5. Why chose those frequencies, such as delta, theta, gamma etc.? What is the rationale behind that?
Please provide scientific evidence by showing that those frequencies fit with your research question in
the introduction. To be concise, the rationale behind the selection of these frequency bands should be
introduced in the introduction section first, so the readers will understand why you chose those
frequencies in the method section.

Specific comments:

1. Please provided specific information about how did you run G*Power. Here is well-written paper in
the participant section for your reference.

Wang, K. P., Cheng, M. Y., Elbanna, H., & Schack, T. (2023). A new EEG neurofeedback training
approach in sports: the effects function-specific instruction of Mu rhythm and visuomotor skill
performance. Frontiers in Psychology, 14, 1273186.

2. My previous question was that what are rested sharshooters meaning? Please clarify it. The authors
stated that Gu et al. showed that in resting state, the duration, occurrence rate, and coverage rate of
microstate D in professional sharpshooters were significantly higher than those in amateur
sharpshooters. The explanation is good. But why the authors did not change it in the introduction
section in the manuscript?

Experimental design

.

Validity of the findings

.

Additional comments

.

Reviewer 3 ·

Basic reporting

The language is greatly improved throughout the manuscript, such that it is clearly understood. The authors have addressed my comments sufficiently in this section, including providing relevant background and definitions in the introduction.

Experimental design

The clarity of the experimental design has been greatly improved, and the research question is clear and defined.

Validity of the findings

The authors have again done well to improve the manuscript based on the reviewer comments. The impact of the results are more clear and easy to follow, and the distinction between cross sectional and acute findings is made more clear. The authors have also included all requested statistical parameters in the results section.

Additional comments

The authors should be commended on improving the quality of English writing in the manuscript, as well as greatly improving the justification and background in the introduction. The methods are clear, and the results have all the necessary statistical parameters reported.

A few minor points below:
Introduction:
• Line 59 “The likelihood of transition from other microstates to microstate D was the highest (F. Gu et al., 2022).” , it is unclear to me what this sentence is referencing. I believe it is a continuation of the sentence above (starting line 57), but a “.” in the middle incorrectly breaks up the sentence.
Methods:
• Do you have any other demographic information to report in Table 1 (not required, but any other measures you collected that could describe the sample would be appreciated).
• Line 114 “priori power analysis indicated that the detection of a large effect with .80 power and .05 Type I error rate would be achievable with 25 participants per group” Do you have justification for expecting a large effect size (please cite the study you based this effect size on)? What was the specific effect size that was used to calculate?

---

## Round 0.3 · Minor Revisions

The manuscript is well-structured, but an additional improvements is necessary as suggested by the reviewer:

Justification for Effect Size: The manuscript includes an expected effect size (effect size f = 0.25) for the power analysis, but it lacks a justification for choosing this specific value. It is essential to explain why 0.25 was selected, citing previous papers, reviews, or meta-analyses investigating the acute effects of exercise on EEG. This rationale is critical to validate the number of subjects used and ensure the study is appropriately powered.

Reviewer 3 ·

Basic reporting

The authors have adequately responded to all my concerns in this category.

Experimental design

Only one point regarding the power analysis. I appreciate that the authors added the expected effect size (effect size f = 0.25) that was used for calculation, and provided details needed to replicate the analysis. However, it is missing a justification for choosing that effect size, why was that specific 0.25 value chosen (i.e. what previous paper or review or meta-analysis investigating the acute effects of exercise on EEG did you base your effect size on?). This is needed to provide a sound rationale for the number of subjects used, and to ensure proper power in the study.

Validity of the findings

The authors have adequately responded to all my concerns in this category.

---

## Round 0.4 · accepted · Accept

I look forward to seeing your future work submitted to PeerJ.